



# Estimating long-term groundwater storage and its controlling factors in cold regions

Soumendra N. Bhanja[1], Xiaokun Zhang[2], Junye Wang[1]

[1]Athabasca River Basin Research Institute (ARBRI), Athabasca University, 1 University Drive, Athabasca, Alberta T9S
3A3, Canada
[2]School of Computing & Information System, Athabasca University, 1 University Drive, Athabasca, Alberta T9S 3A3,
Canada

*Correspondence to*: Junye Wang (junyew@athabascau.ca)

**Abstract.** Groundwater is one of the most important natural resources for economic development and environmental
sustainability. However, groundwater storage can be significantly affected by climate change through permafrost thaw,
snowpack change, and glacier retreat in cold climate regions, and human activities due to over-use and over-extraction of
resources. Therefore, it is very important to be able to estimate long-term groundwater storage for biodiversity and
sustainable development. In this study, we estimated groundwater storage in 11 major river basins across Alberta, Canada
using a combination of remote sensing (Gravity Recovery and Climate Experiment-GRACE), in situ surface water data, and
land surface modelling estimates ($GWSA_{sat}$). We applied separate calculations for unconfined and confined aquifers, for the
first time, to represent their hydrogeological differences. Storage coefficients for the individual wells were incorporated to
compute the monthly $GWSA_{obs}$. The $GWSA_{sat}$ from the two satellite-based products were compared with in situ groundwater
storage ($GWSA_{obs}$) estimates. The estimates of $GWSA_{sat}$ were in good agreement with the $GWSA_{obs}$ in terms of pattern and
magnitude (e.g., RMSE ranged from 2 to 14 cm). While comparing $GWSA_{sat}$ with $GWSA_{sat}$, most of the statistical analyses
provide mixed responses, however the Hodrick-Presscott trend analysis clearly showed a better performance of the GRACE-
mascon estimate. The results showed trends of $GWSA_{obs}$ depletion in 5 of the 11 basins. Our results indicate that the
precipitation played an important role in influencing the $GWSA_{obs}$ variation in 4 of the 11 basins studied. Water budget
analysis showed an availability of comparatively lower terrestrial water in 9 of the 11 basins in the study period. Historical
groundwater recharge estimates indicate a reduction of groundwater recharge in 8 basins during 1960-2009. The output of
this study could be used to develop sustainable water withdrawal strategies in Alberta, Canada.

## 1 Introduction

Fresh water is an important resource for economic development and social sustainability around the world. Approximately,
1.2 billion people live in water scarce areas across the globe (UN-Water/FAO, 2007). More than a billion people lack access
to safe drinking water and this number is increasing due to an increasing population (Connor, 2015). However, the effects of
climate change on glaciers and snowpack, and human activities, such as over-use and over-extraction of resources, can result



in lowering water tables and groundwater depletion (Scanlon et al., 2016; Bhanja et al., 2017b). It is reported that groundwater systems are vulnerable in some of the densely populated regions across the globe (Rodell et al., 2009; Tiwari et al., 2009; Wada et al., 2010; Scanlon et al., 2012; Bhanja et al., 2014; Richey et al., 2015; Chen et al., 2016; Bhanja et al., 2017b). Intense anthropogenic groundwater withdrawal could lead to groundwater drought in countries such as India

(Goldin, 2016). Furthermore, groundwater over-use and excessive pumping could lead to severe water contamination due to saltwater intrusion (Mukherjee et al., 2015; MacDonald et al., 2016) and arsenic mobility (Harvey et al., 2002; Fendorf et al., 2010).

In particular, groundwater flow in permafrost regions is restricted by the ice-rich soil layers at depth, which acts as an aquiclude (Woo, 1986).Groundwater storage might be affected by the effects of climate change on the snowpack and glaciers

in arid, cold climate regions, such as Alberta, Canada. For example, a significant portion of the population (approx. 20% or 600,000 people) depends on groundwater for their domestic use in Alberta (AEP, 2011).  Total licensed groundwater use in Alberta was estimated to be 300 million cubic meters (MCM) in 2010 (AEP, 2011). Out of the total groundwater usage, the industrial sector accounted for 41%, followed by agriculture and domestic sectors at a rate of 23% and 19% respectively (AEP, 2011). An increased level of occurrences of Arsenic in groundwater have been reported in the Cold Lake region of

Alberta, as outcome result of human activity (Javed and Siddique, 2016). Therefore, groundwater resources must be properly managed to balance the needs of domestic, industrial and irrigational water for sustainable development (Konikow and Kendy, 2005). To achieve a balance of groundwater resources, it is necessary to estimate long-term groundwater storage and supply at the regional scale.

In situ monitoring of wells is the traditional approach for estimating groundwater storage. However, well monitoring is

spatially not continuous and has a high cost at a large region scale., There are only scant observation stations in some areas, especially in semi-arid and arid environments, or cold climate regions covered by glacier and snowpack, due to difficulties of access and monitoring. As a result, proper groundwater management and decision-making are hampered considerably by the scarcity of data. Remote sensing data from the Gravity Recovery and Climate Experiment (GRACE) satellite mission could be used to estimate groundwater storage at a continuous and large scale across the globe, and offers a new opportunity for

groundwater storage assessment (Rodell et al., 2007). Although the GRACE satellite mission currently provides global-scale data for the detection of temporal gravity changes (Tapley et al., 2004), these temporal gravity changes are not a direct measurement of groundwater storage. A relationship would have to be established between temporal gravity changes and groundwater storage variations through the continuously evolving algorithms (Watkins et al., 2015). Estimates of groundwater storage using the remote sensing have been  performed  around the globe (Swenson et al., 2006; Rodell et al.,

2007; Strassberg et al., 2007; Scanlon et al., 2012; Shamsudduha et al., 2012; Voss et al., 2013; Panda and Wahr, 2016; Bhanja et al., 2016; Long et al., 2016; Bhanja et al., 2017b). Huang et al. (2016) used remote sensing data for computing the groundwater storage anomalies (GWSA) in order to estimate groundwater storage in Alberta. In their study, the ground water levels at 36 wells, mostly confined to the southern Alberta region, were correlated with both the GRACE total water storage (TWS) and GWS variations. Then they compared the TWS with groundwater levels instead of the groundwater



storage without considering surface water data due to the lack of available high resolution data. Recent studies (e.g. Huang et al., 2015; Nanteza et al., 2017) have considered both confined and unconfined aquifers for in situ GWSA computation but they have not separated the data from the two types. Rodell et al. (2007) indicated the importance of surface water factors in the GWSA estimation and sought for inclusion of surface water storage variations in GWSA disaggregation. They also

pointed out the importance of separating contributions to temporal mass variability using auxiliary observations and numerical models when estimating groundwater storage changes in large scale regions. In cold climate regions, such as in Alberta, the surface water could make a significant contribution to groundwater storage variations due to the effects of climate change on snowpack, glaciers, permafrost, and wetlands. Therefore, more efforts are required to properly evaluate groundwater storage for specific yield values in transforming groundwater level information to groundwater storage in cold

climate regions (Feng et al., 2013).

The main objectives of this study are:

1. To investigate the long term groundwater storage conditions in cold climate regions, such as the 11 river basins in Alberta, by combining all of the processing steps, such as the surface water storage estimates.

2. To validate the remote sensing estimates from two different remote sensing products using the maximum available

in situ observation well data. The in situ groundwater storage has been estimated by combining the storage coefficients and aquifer thickness (for confined aquifers) with the water table fluctuation.

To find the role of natural hydrological components (e.g. precipitation, evapotranspiration) in groundwater storage variations in the studied basins. We have studied long-term groundwater recharge trends from a global-scale hydrological model for inferring long-term variabilities in groundwater recharge rates.

**2 Materials and Methods**

**2.1 Study area**

This study has been conducted in the major river basins (the map has been made following Lemay and Guha, 2009; AEP, 2011; AEP, 2017) within the province of Alberta (Figure 1). The Peace River basin is the largest basin in the province, followed by the Athabasca River basin and Hay River basin, respectively (Table 1). Basin-scale, annual average

precipitation varies within 330 to 570 mm/year (Table 1). We used Global Land Cover Facility (GLCF) native resolution data (resolution: ~460 m × 460 m; www.landcover.org) for characterizing land cover (Channan et al., 2014). Most of the land areas in Alberta are covered by natural vegetation (i.e. Forest, Shrubland, mixture of Shrub and Grassland (MSG), and Grassland; Figure 1c, Supplementary Table 1). The second most prevalent land-cover type is cropland (Figure 1c, Supplementary Table 1). Surface water bodies (water and permanent wetland) cover less than 6% of the area of all the river

basins (Figure 1c, Supplementary Table 1).

We used monthly mean precipitation data from the archives of the Climatic Research Unit (CRU), University of East Anglia. The quality controlled, gridded $0.25 \times 0.25^0$, monthly mean TS4.0 total precipitation products are used here (Harris et al.,





2014). The precipitation-gauge based data was collected through the World Meteorological Organization (WMO), National Oceanographic and Atmospheric Administration (NOAA), as well as other international and national agencies across the globe for preparing this dataset (Harris et al., 2014). The precipitation data are spatially averaged in order to provide basin-scale data. CRU data have been found to have the best match of other available products while comparing with in situ

precipitation measurements in China (Zhao and Fu, 2006). Precipitation data exhibits temporal as well as spatial variations in the study period with values of 150 mm/year to >1000 mm/year (Figure 2). In general, the lowest precipitation has been observed in 2004 and the highest in 2010 (Figure 2). Spatially averaged basin-scale precipitation values indicate the highest precipitation rates prevail in the North Saskatchewan River basin (basin 4, 573 mm/year; Table 1) and lowest rates prevail in the South Saskatchewan River basin (basin 9, 334 mm/year; Table 1). Precipitation rates are highly seasonal in Alberta

(Figure 3).

**2.2 In situ measurements of groundwater storage**

Groundwater level (GWL) depth data are obtained from the Government of Alberta, Alberta Environment and Parks (http://environment.alberta.ca/apps/GOWN/#). Daily GWL depth data are obtained for 470 monitoring wells distributed across the province of Alberta. The data is screened for data continuity (at least 80% of the data are present in each location)

within the study period 2003-2015, resulting in the use of GWL data from 157 measurement locations. Daily GWL information is converted to monthly GWL at individual locations. Because of the differences in GWS variations within different types of aquifers, these wells need to be classified as unconfined, semi-confined and confined. Out of the 157 measurement locations used in the study, 24 are located in unconfined aquifers, 17 are located within semi-confined aquifers, 100 are located within confined aquifers and 16 are unclassified.

We have studied the subsurface hydrogeology in detail using well-specific lithology information from the Government of Alberta, Alberta Environment and Parks (http://environment.alberta.ca/apps/GOWN/#). The wells located in unclassified or semi-confined aquifers are characterized as either confined or unconfined based on their location hydrogeology and screen depth. In order to compute groundwater storage anomalies ($GWSA_{obs}$) in an unconfined aquifer, the GWSA needs to be accurately represented using the storage coefficients of groundwater levels (Scanlon et al., 2012). We have followed the

equation (Todd and Mays, 2005; Bhanja et al., 2017a):

$$GWSA_{obs} = (h_m \times S_y - h_i \times S_y) \tag{1}$$

Where, $h_m$ and $h_i$ represent the mean GWL depth and GWL depths at different time periods at a location; $S_y$ represents the specific yield of the aquifer. $S_y$ is assigned to the individual data based on the specific yield of the geologic material in its screen position. Specific yield data corresponding to a specific geologic material are presented in the Supplementary Table 2.

$GWSA_{obs}$ in a confined aquifer have been estimated following the equation (Todd and Mays, 2005):

$$GWSA_{obs} = (h_m \times S_s \times b - h_i \times S_s \times b) \tag{2}$$

Where, $S_s$ is the specific storage and b is the thickness of the aquifer. $S_s$ of a material varies over a wide range, the details of material-specific $S_s$ are provided in Supplementary Table 3. Thickness of the aquifer for the individual aquifer units is





obtained from the Government of Alberta, Alberta Environment and Parks. The data are assigned to individual wells based on their screening zone and thickness of the particular aquifer unit.

### 2.3 Surface water storage processing

Surface water level daily time-series are obtained (n= 393) from the Water Office, Government of Canada
(wateroffice.ec.gc.ca) for the study region. After rearranging the data based on near-continuous data availability, we used 65 locations with > 80% of the data availability range. The data are temporally averaged at each location for estimating monthly mean values. The number of locations that fall within each of the river basins are spatially averaged to obtain the month-scale spatially averaged surface water anomaly. The surface water coverage fraction varies over the study region (Figure 1c and Supplementary Table 1). In order to get realistic surface water storage variations, surface water area fractions have been
multiplied with the spatially averaged surface water anomaly in each river basin.

### 2.4 Gravity recovery and climate experiment (GRACE)

We have obtained the monthly mean liquid water equivalent thickness, $1^0 \times 1^0$ gridded data from the archives of the National Aeronautics and Space Administration's (NASA) Jet Propulsion Laboratory (JPL) archive. JPL-mascon solutions, version RL05, was used for 137 months (the data were not available for some of the months, details can be found in Watkins et al.,
2015) between January 2003 and April 2015. The GRACE mission observes changes in gravity in the Earth's subsurface and provide the data on a continuous basis. The gravity change information has been processed further in order to obtain the terrestrial water storage (TWS) change data (details can be found here: http://grace.jpl.nasa.gov/data/get-data/jpl_global_mascons/ accessed on 14 November, 2017). Satellite Laser Ranging (SLR) has been incorporated for estimating degree 2 and order 0 coefficients (Cheng and Tapley, 2004). Processes to improve the geocenter correction have
been reported by Swenson et al. (2008). The Glacial isostatic adjustment (GIA) related post glacial rebound signals are removed using the process by Geruo et al. (2013). In the mascon approach, the entire globe is characterized as equally-spaced $3^0$ spherical mass concentration blocks (Watkins et al., 2015). In order to improve the TWS estimates, scale factors provided with the data are multiplied (Bhanja et al., 2016). Scale factors are estimated in order to improve the performance of the TWS estimates.

The spherical harmonics (SH) related TWS information has been obtained for 137 months (between January 2003 and April 2015) from the NASA JPL archive. We used $1^0 \times 1^0$ gridded RL05 data sets of SH solutions (Landerer and Swenson, 2012). Three independent solutions from the Center for Space Research at the University of Texas at Austin, the NASA JPL, and the German Space Agency (GFZ), were retrieved and combined to use in this study. Like the mascon approach, several similar techniques are applied to obtain the TWS change in the SH approach (Source: http://grace.jpl.nasa.gov/data/get-
data/monthly-mass-grids-land/ accessed on 14 November, 2017). Errors associated with N-S stripes in the TWS data are removed using a destriping filter. A Gaussian filter at 300-km width is also applied to the data. In order to improve the TWS estimates, the scale factor provided with the data are multiplied (Bhanja et al., 2016).





One advantage of the mascon approach is the introduction of *a priori* information that leads to the removal of correlated noise (stripes) in the data. As a result, post-processing filters are not required to be applied (Watkins et al., 2015). TWS data obtained from the mascon approach are less dependent on scale factors for estimating basin-scale mass change estimates (Watkins et al., 2015).

**2.5 Estimating groundwater storage from remote sensing and global models**

Satellite-based groundwater storage anomalies ($GWSA_{sat}$) are estimated using a mass balance approach after removing other components of the hydrological cycle from the TWSA. These components include soil moisture anomaly (SMA), anomalies in snow water equivalents (SNA), and anomalies in surface water variations (SWA). Anomalies are estimated after removing the all-time mean value from the individual monthly values for all of the components. Soil moisture and snow water equivalents data were retrieved from NASA's Global Land Data Assimilation System (GLDAS) (Rodell et al., 2004) for 148 months in the study period. Bhanja et al. (2016) reported better GWSA estimates while using a combination of data from simulations of three different land surface models (LSM), comparing the use of any single model's output. We also used a combined estimate from the outputs of the Community Land Model (CLM), Variable Infiltration Capacity (VIC), and Noah (Rodell et al., 2004). Surface water variation plays an important role in estimating GWSA. We have computed the surface water variations using in situ data, described in section 2.3. GWSA can be estimated using the following equation:

$$GWSA_{sat} = TWSA - SMA - SNA - SWA \qquad (3)$$

**2.6 Statistical approaches**

In order to compare the datasets using statistically robust techniques, we have used the root mean square error (RMSE), Pearson's correlation, skewness, kurtosis, and the coefficient of variation. RMSE has been used to show the departures from the true (in situ estimates here) value (Helsel and Hirsch, 2002).

$$RMSE = \sqrt{\sum_{i=1}^{n} \frac{(x_i - y_i)^2}{n}} \qquad (4)$$

$$Pearson's\ correlation\ (r) = \frac{\sum_{i=1}^{n}(x_i - \bar{x})(y_i - \bar{y})}{\sqrt{\sum_{i=1}^{n}(x_i - \bar{x})^2}\sqrt{\sum_{i=1}^{n}(y_i - \bar{y})^2}} \qquad (5)$$

$$Skewness = \frac{\frac{1}{n}\sum_{i=1}^{n}(x_i - \bar{x})^3}{\sqrt{\frac{1}{n-1}\sum_{i=1}^{n}(x_i - \bar{x})^2}^3} \qquad (6)$$

$$Kurtosis = \left\{ \frac{n(n+1)}{(n-1)(n-2)(n-3)} \sum_{i=1}^{n} \frac{(x_i - \bar{x})^4}{\sigma} \right\} - \frac{3(n-1)^2}{(n-2)(n-3)} \qquad (7)$$

$$Coefficient\ of\ variation\ (CV) = \frac{\sigma}{\mu} \qquad (8)$$

Where, $x_i$ and $y_i$ are the two different GWSA estimates i.e. $GWSA_{obs}$ and $GWSA_{sat}$, $i$ (= 1, 2, 3, …, $n$) is the number of samples; $\bar{x}$ and $\mu$ indicate mean values; $\sigma$ indicate standard deviation.





The trend analyses are provided from linear regression analysis. In order to represent the non-linearity present within the data, we used the Hodrick-Prescott (HP) filter (Hodrick and Prescott, 1997), a non-parametric, non-linear trend analysis. The HP filter employs a specific approach for separating trend ($T_t$) and cycle ($c_t$) components in the data ($y_t$).

$$y_t = T_t + c_t \tag{4}$$

In order to estimate the trend and cycle separately, the HP filter solves the following equation (Hodrick and Prescott, 1997):

$$\text{Min (T)} \sum_{t=1}^{T}((y_t - T_t)^2 + \lambda((T_{t+1} - T_t) - (T_t - T_{t-1}))^2 \tag{5}$$

Where $T_{t+1}$ and $T_{t-1}$ represents the trend component with time steps of *t+1* and *t-1*, respectively. The long term average of the cyclical components is close to zero (Hodrick and Prescott, 1997). The smoothing parameter ($\lambda$) is a positive number that reduces the variability within cyclical components (Hodrick and Prescott, 1997). The value of $\lambda$ was chosen to be 14400 for

monthly data (Hodrick and Prescott, 1997; Ravn and Uhlig, 2002).

## 3 Results and Discussions

### 3.1 Groundwater storage anomalies

In situ GWSA (GWSA$_{obs}$) values ranged from -30 to 30 cm in all of the basins, with the highest fluctuations observed in basin 7. GWSA$_{obs}$ exhibits near zero values in basins 5, 6, 10 and 11 (Figure 3). GWSA$_{obs}$ magnitudes in different basins can

arise as a result of diversity in specific yield values in the underlying material (Supplementary Table 2). In situ estimates show seasonality, i.e. variations with precipitation rates etc., in basin 7. Trends in the GWSA$_{obs}$ show decreasing GWSA between 2003 and 2015 in basins 2, 3, 7, 8 and 9 (Table 1). The results indicate that GWSA$_{obs}$ depletion is in the range of -0.20 in basins 2 and 3. It is interesting to note that the basins 7, 8 and 9 are composed of >25% cropland (Supplementary Table 1). On the other hand, basin 3 has been subjected to the highest amount of licensed groundwater withdrawal allocation

in Alberta (basin 3 accounts for 39% of the total groundwater usage in Alberta). On the other hand, an increasing trend has been observed in the remaining basins (Table 1). One probable cause for the groundwater table increase in these basins could be related to precipitation variability. The study region has been subjected to large-scale drought during 1999-2005 (Hanesiak et al., 2011). As a result, the TWS recovery in 2004-2007 has also been observed by Lambert et al. (2013).

Remote sensing estimates of GWSA (GWSA$_{sat}$) using the two different estimates, GRACE-MS and GRACE-SH approaches,

show temporal variations ranging from -20 to 20 cm. However, the seasonal amplitudes are not similar in different basins (Figure 3). In general, the magnitude of the GWSA$_{sat}$ compares well with that of the GWSA$_{obs}$ (Figure 3). GWSA$_{sat}$ exhibits a large amplitude in basins 4, 7 and 8 (Figure 3). In general, the GWSA$_{sat}$ estimates from the two products match well (Figure 3). The estimations are in line with the values reported for the Mackenzie River basin in Scanlon et al. (2018).





### 3.2 RMSE, correlation, skewness and kurtosis

Deviations from the observed values are measured by the root mean squared error (RMSE) that combines both bias and lack of precision (Helsel and Hirsch, 2002). The RMSE estimates show a good match of satellite-based GWSA estimates in comparing the in situ estimates. RMSE was found to be within 5 cm in most of the basins (Figure 4a). In general, both of the

satellite-based estimates exhibit similar RMSE in basins 2, 3, 5, 6 and 11 (Figure 4a). Pearson's correlation coefficient (r) provides information on the linear association between the two variables (Helsel and Hirsch, 2002). Comparing the two products, correlation coefficients are found to be higher for the MS product in most of the basins (Figure 4b). Skewness has been used to represent the symmetricity in the data distribution (Helsel and Hirsch, 2002) and kurtosis has been used to represent the tail length of data distribution. Skewness and kurtosis have been used here in order to compare the GWSA

estimates from the two satellite-based estimates with the in situ estimate. Comparing the two estimates, skewness and kurtosis analyses provide mixed results. For example, one product provides better results in some of the basins, the other in the remaining basins (Figure 4c, 4d).

### 3.3 Coefficient of variation (CV) and scatter analyses

Data dispersion can be measured through the coefficients of variation (CV). In general, CV data are found to match well for

the two satellite-based products and the in situ estimates (Figure 5). CV data shows mixed responses when comparing the two satellite-based estimates. Scatter analysis shows the characteristic of the relationship (i.e. linear, non-linear) between the two variables (Helsel and Hirsch, 2002). Scatter analysis results do not provide any distinct comparison between the MS, SH estimates and the in situ estimate (Figure 6). The in situ data contains signatures of individual wells and, as a result, are influenced by local-scale climatic, hydrogeologic, and anthropogenic responses. However, the satellite-based estimates are

providing responses from a large region and may not be influenced by local-scale fluctuations (Bhanja et al., 2016).

### 3.4 Hodrick-Prescott (HP) filter estimates

We used a non-parametric filtering (HP filter) approach for computing the trends in GWSA and compared the results with in situ estimates. HP trends of GWSA$_{obs}$ show the recent depleting trends in basins 1, 2, 3, 7 and 9 (Figure 7). In general, the HP trends of satellite-based estimates are relatively similar to each other. However, a comparatively better match of GWSA

for the GRACE-MS product and in situ estimates has been observed in basins 4, 5, 6, 10 and 11. Significantly negative (*p value* <0.01) correlation has been observed for both estimates in basins 7, 8 and 9, which are subjected to irrigation with >25% land area coverage affected (Figure 1b and Supplementary Table 1).

### 3.5 GWSA and precipitation relationship

In general, precipitation is the major controlling factor for variations in water storage (Scanlon et al., 2012). In this study, we

have observed that GWSA values are not directly influenced by the precipitation pattern in some of the basins. The HP trend



analysis shows a good match of GWSA$_{obs}$ with precipitation in basins 1 and 10 only (Supplementary Table 4). The cross-correlation analysis provide similar inferences (Supplementary Table 4). In order to investigate the relationship with more detail, the Granger causality analyses (Granger, 1988) were performed with order 1 (insignificant results were found when other orders were used). Results show precipitation significantly (*p value* <0.01) causes GWSA in 4 of the 11 studied basins,

basin 1, 5, 7 and 11. The results were found to be insignificant or even negatively correlative in other basins (Supplementary Table 4). This implies that other factors, such as domestic and industrial water withdrawal, recharge from snowmelt etc., play a major role in influencing the GWSA in those basins.

### 3.6 GWSA and the natural water budget

Observation of a non-significant relationship of precipitation and GWS anomalies in most of the basins indicated the

influence of other factors controlling GWS. The natural water availability for terrestrial water components (i.e. groundwater, surface water, soil moisture, etc.) have been studied by delineating the difference (DIFF) between precipitation (P) and evapotranspiration (ET) in another way, called the net precipitation flux (Syed et al., 2005; Rodell et al., 2015). Long et al. (2014) found ET estimates from LSMs provide the best ET estimates comparing in situ observations. We retrieved data from the simulation of the Noah land surface model, version 2.1, as a part of the GLDAS simulation (Rodell et al., 2004). The

GLDAS include observation data from satellite sensors and ground-based measurements in order to improve the simulation output (Rodell et al., 2015). The DIFF data exhibit negative values during summer months (Figure 9). Comparatively lower P and higher ET values are observed in the summer months, making the DIFF negative. The basin-wise DIFF values show reducing estimates in 9 of the 11 basins with the highest being observed in the Peace River basin (basin 2), where the DIFF estimate shows a net reduction of 0.41 km$^3$ of water between 2003 and 2015 (Table 1). Reduction in DIFF values is putting

stress on terrestrial water as well as groundwater conditions in the study region. In order to find the historical groundwater recharge pattern, we used long-term (1960-2009) groundwater recharge data from a global-scale hydrological model, WaterGAP (details of WaterGAP simulation can be found in Doll et al., 2014). We used a combination of diffuse groundwater recharge and recharge from the surface water bodies, which we termed as "total groundwater recharge". The simulated, historical total groundwater recharge was found to be negative in 8 out of 11 basins, suggesting a change in

rechargeable water volume. Groundwater storage, being a combination of recharge from precipitation and surface water bodies, the inter-aquifer flow, discharge to surface water bodies and the anthropogenic withdrawal, could be largely impacted from reductions of the first two terms. Increasing human activities linked with groundwater withdrawal could lead to severe groundwater stress if it continues uncontrolled.

### 3.7 Assumptions and limitations

On the basis of data availability, we have not included the entire extent of the river basins in the current analysis. The river basins are selected based only on their geopolitical extent in the province of Alberta (Figure 1). For in situ estimates, GWSA information is spatially averaged for providing the basin's GWSA estimates and also to compare them with the satellite-



based estimates following Bhanja et al. (2017a) and Scanlon et al. (2018) etc. The time-period of the study is restricted by the availability of data. Separation of GWSA signals from TWSA by removing all other components is a challenging task due to the lack of in situ measurements of other components and the large uncertainties associated with LSM simulated products (Scanlon et al., 2015).

**4 Conclusions**

A network of 470 daily groundwater monitored wells was used to compute groundwater storage anomalies (GWSA) in 11 major river basins in Alberta, Canada between January 2003 and April 2015. Well-specific hydrogeology information and separate treatment of the unconfined and confined aquifers were used for the calculation. Results show that the GWSA trends exhibit depletion in some of the basins that are dominated by anthropogenic groundwater withdrawal, either from

irrigational use or domestic and industrial uses. A GWSA depletion rate has been observed as high as -0.20 cm/year in the Athabasca River basin. The GWSA estimates obtained from remote-sensing probes provided opportunities to evaluate groundwater conditions in remote, ungauged regions. We used two recently released satellite products for estimating $GWSA_{sat}$ in the studied basins. A combination of surface water measurements (n=393) and land surface model estimates of soil moisture and snow water equivalents were used. In general, our remote sensing estimates are in good agreement with

that of the observed estimates, implying that remote sensing estimates could be used in future to monitor groundwater storage in the region at a near-continuous rate. We have investigated the influence of precipitation on GWSA variations. Results show that precipitation caused significant GWSA variations in 4 out of 11 studied basins, indicating prevalence of other factors for influencing GWSA in the remaining basins. Water budget analysis of terrestrial water availability show reductions of available water during the study period in 9 basins. Results indicate groundwater recharge rates have been

decreasing from 1960-2009 in 8 of the basins studied. Outputs of this study may be used to frame sustainable water withdrawal strategies in Alberta, keeping in mind the available water for groundwater recharge.

**Acknowledgements and Data**

The authors would like to thank the Alberta Economic Development and Trade for the Campus Alberta Innovates Program Research Chair (No. RCP-12-001-BCAIP). We would also like to thank Mr. Jim Sellers for the proofreading. This study

uses open-source data from the Groundwater Observation Well Network (GOWN), Alberta Environment and Perks. Surface water data were obtained from the Water Office, Government of Canada (wateroffice.ec.gc.ca). Climatic Research Unit's (CRU) precipitation data were obtained from the CRU achieves from the University of East Anglia. GRACE land data were processed by Sean Swenson, supported by the NASA MEaSUREs Program, and is available at http://grace.jpl.nasa.gov. The GLDAS data used in this study were acquired as part of the mission of NASA's Earth Science Division and archived and




distributed by the Goddard Earth Sciences (GES) Data and Information Services Center (DISC). The WaterGAP model outputs are retrieved from the University of Frankfurt archive: https://www.uni-frankfurt.de/45217892/datensaetze.

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


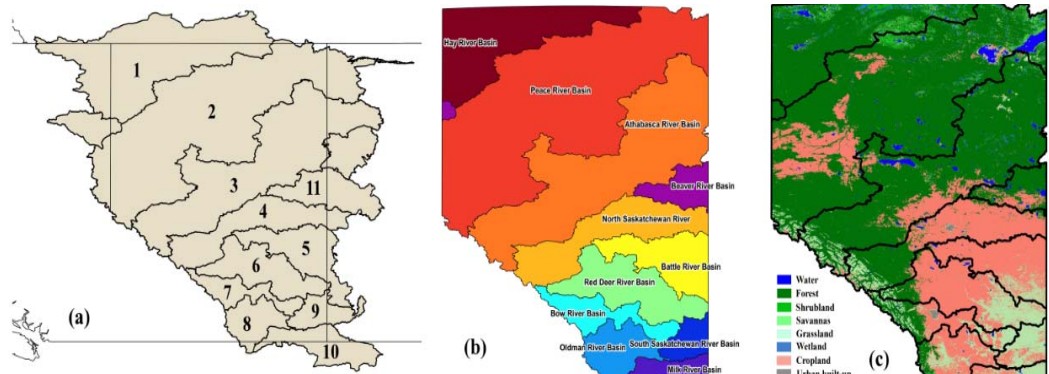

**Figure 1: Major river basins in Alberta, (a) full basin extent; (b) Alberta only; (c) dominant land cover types in Alberta, overlaid by basin boundaries**




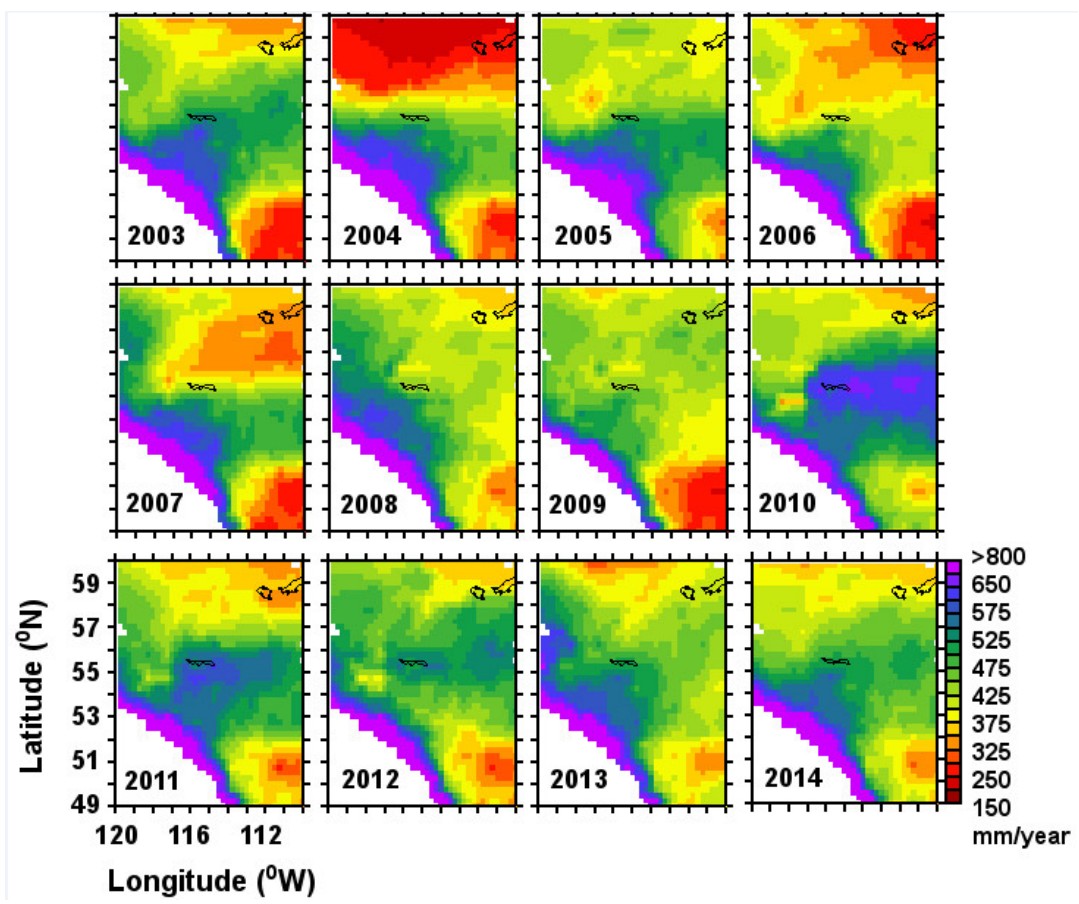

**Figure 2: Annual precipitation rates (mm/year) in Alberta between 2003 and 2014**





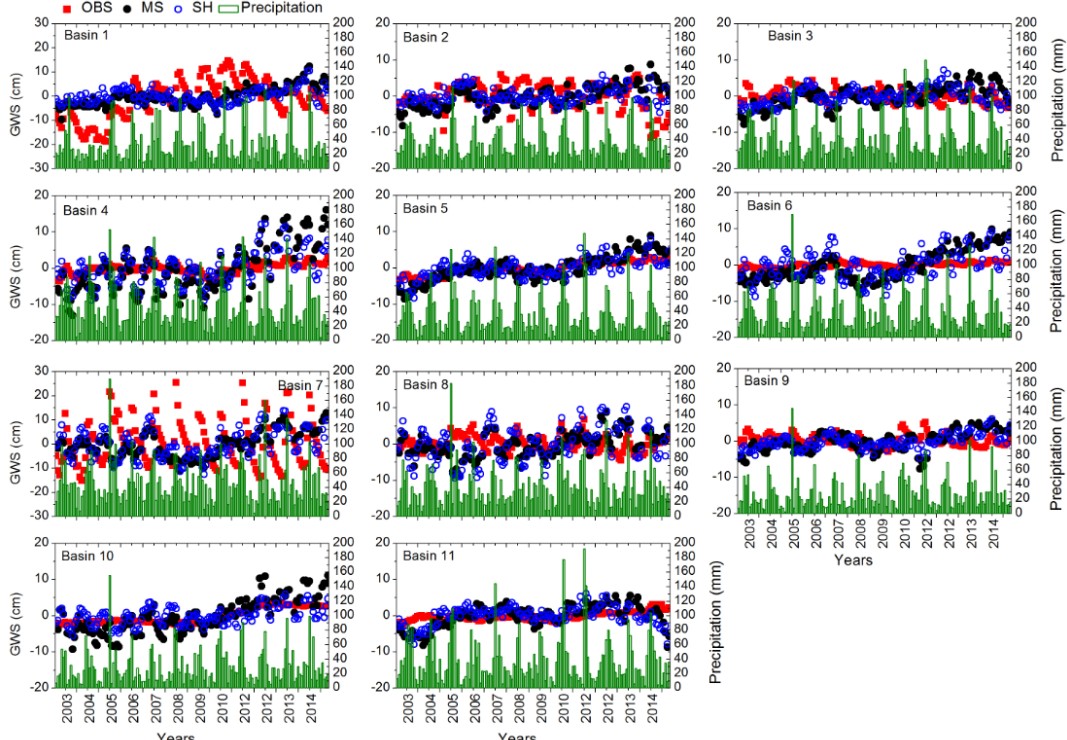

**Figure 3: Basin-wise, monthly time-series of in situ GWSA (OBS, red squares), GWSA obtained using GRACE mascon product (MS, black filled circles) and GRACE SH products (SH, blue open circles), respectively. Monthly, spatial averaged precipitation data are shown using green columns**





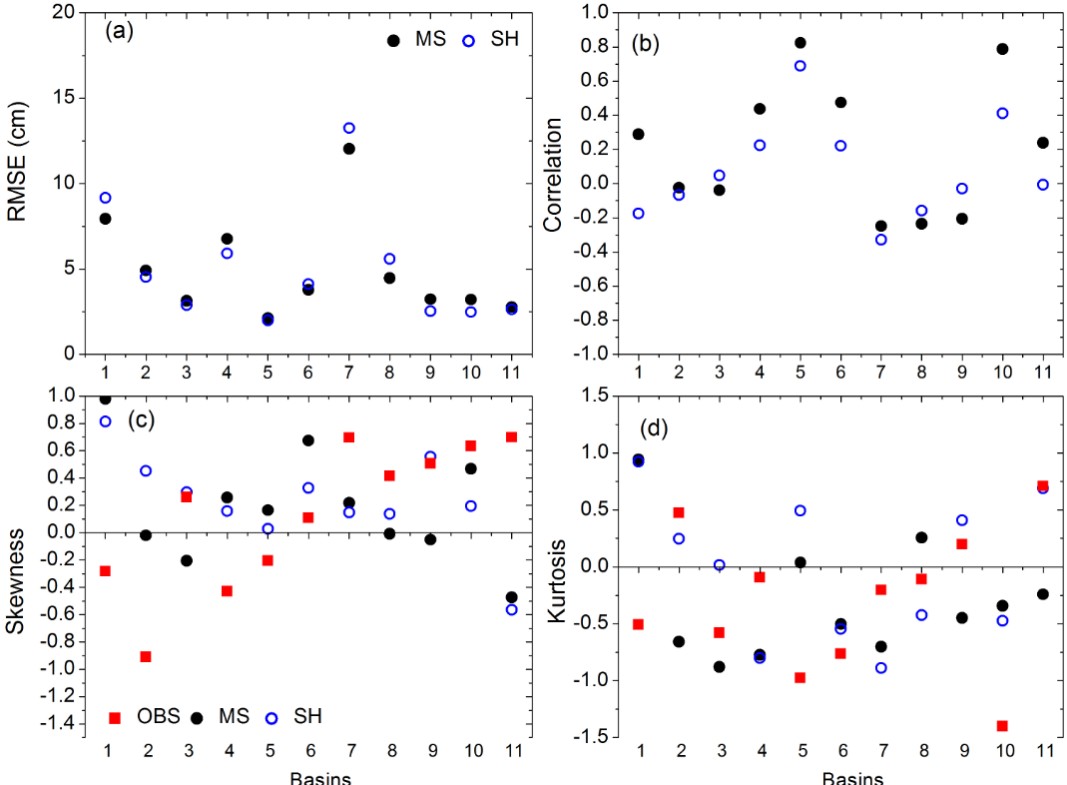

**Figure 4: Basin-wise estimates of (a) RMSE; (b) correlation; (c) skewness and (d) kurtosis**





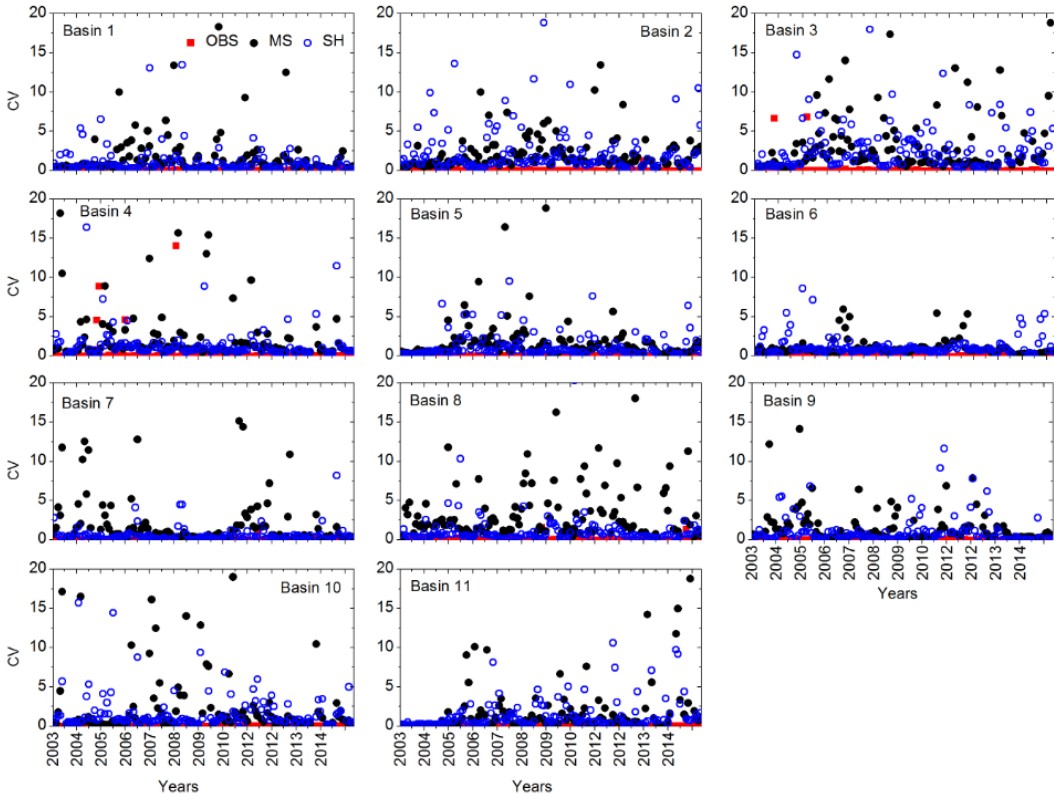

**Figure 5: Basin-wise coefficient of variation (CV) analysis for in situ GWSA (OBS, red squares), GWSA obtained using GRACE mascon product (MS, black filled circles) and GRACE SH products (SH, blue open circles), respectively**



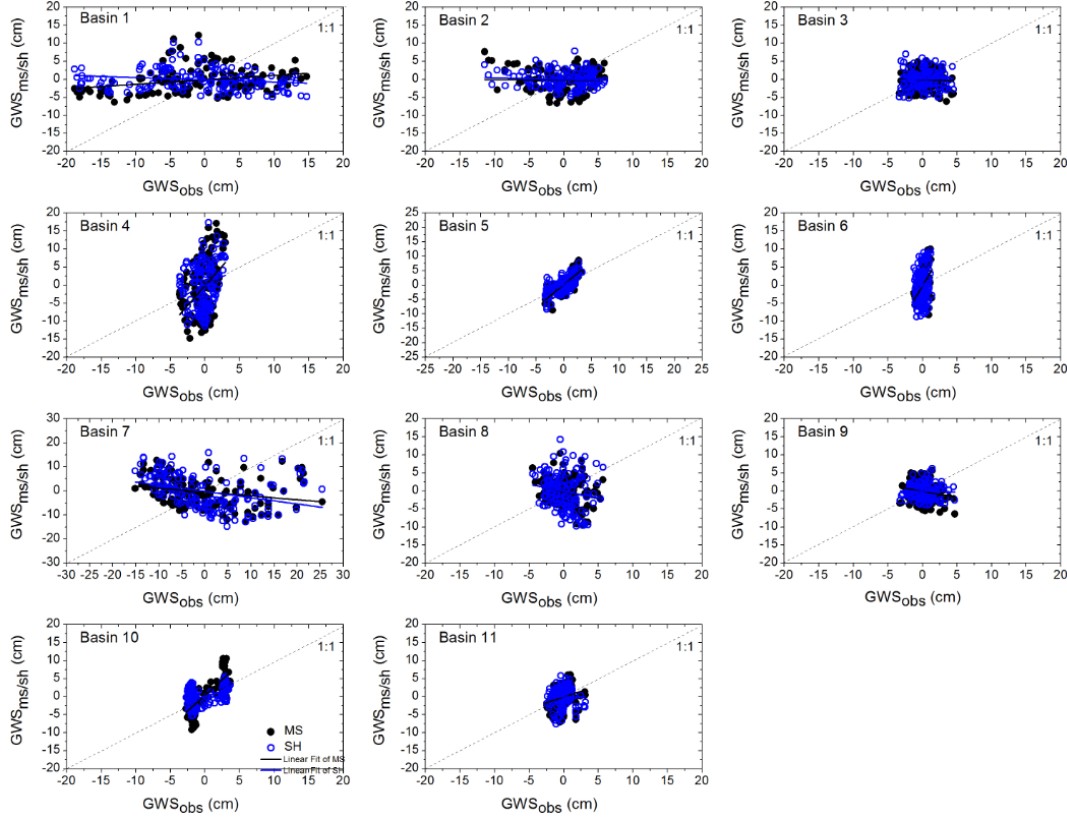

**Figure 6: Basin-wise scatter analysis of in situ GWSA with the GWSA obtained using GRACE mascon product (MS, black filled circles) and GRACE SH products (SH, blue open circles), respectively**



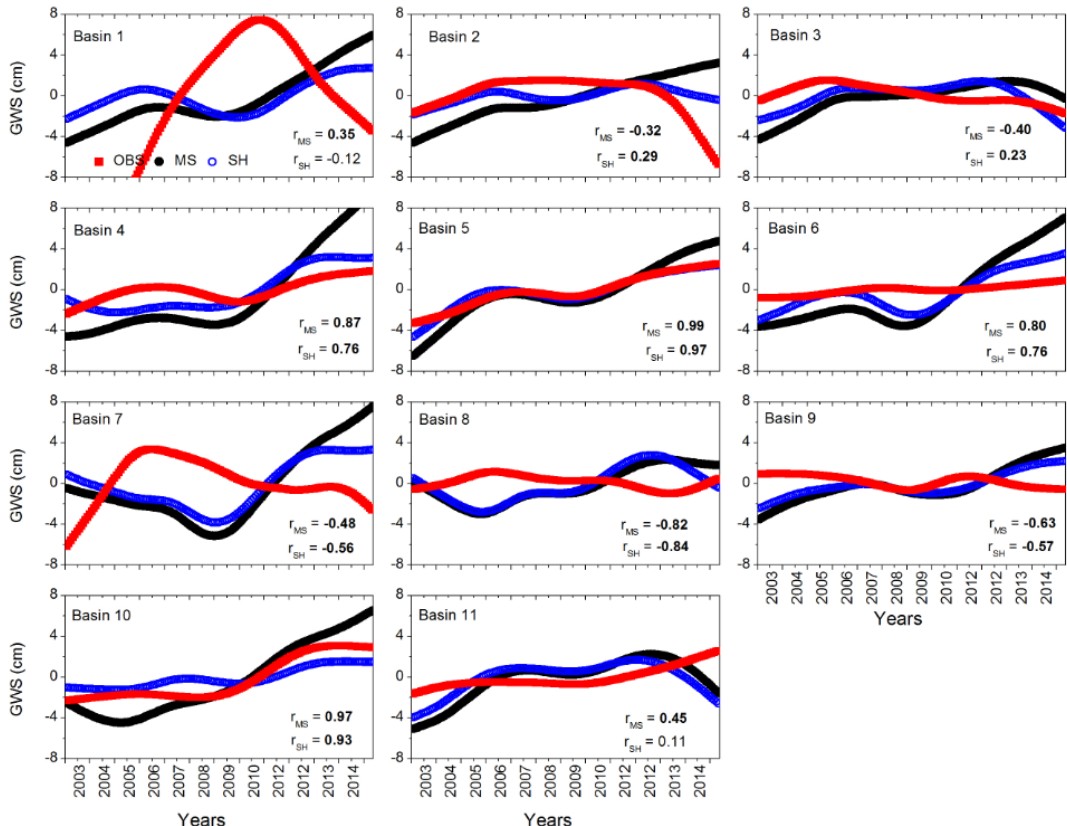

**Figure 7: Basin-wise time-series of HP filter data for in situ GWSA (OBS, red squares), GWSA obtained using GRACE mascon product (MS, black filled circles) and GRACE SH products (SH, blue open circles), respectively. Pearson's correlation coefficient values are provided in in-set and statistically significant (*p value* < 0.01) values are shown in bold font**



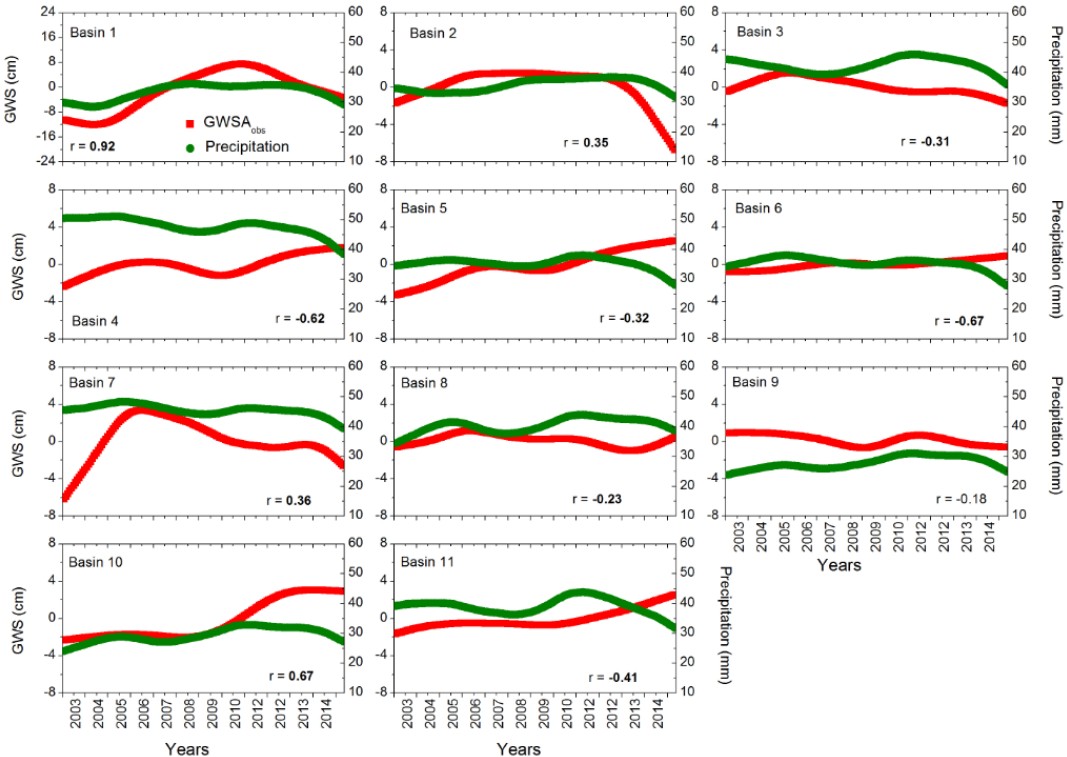

**Figure 8: Basin-wise time-series of HP filter data for in situ GWSA (OBS, red squares) and precipitation data (green circles). Pearson's correlation coefficient values are provided in in-set and statistically significant (*p value* < 0.01) values are shown in bold font**





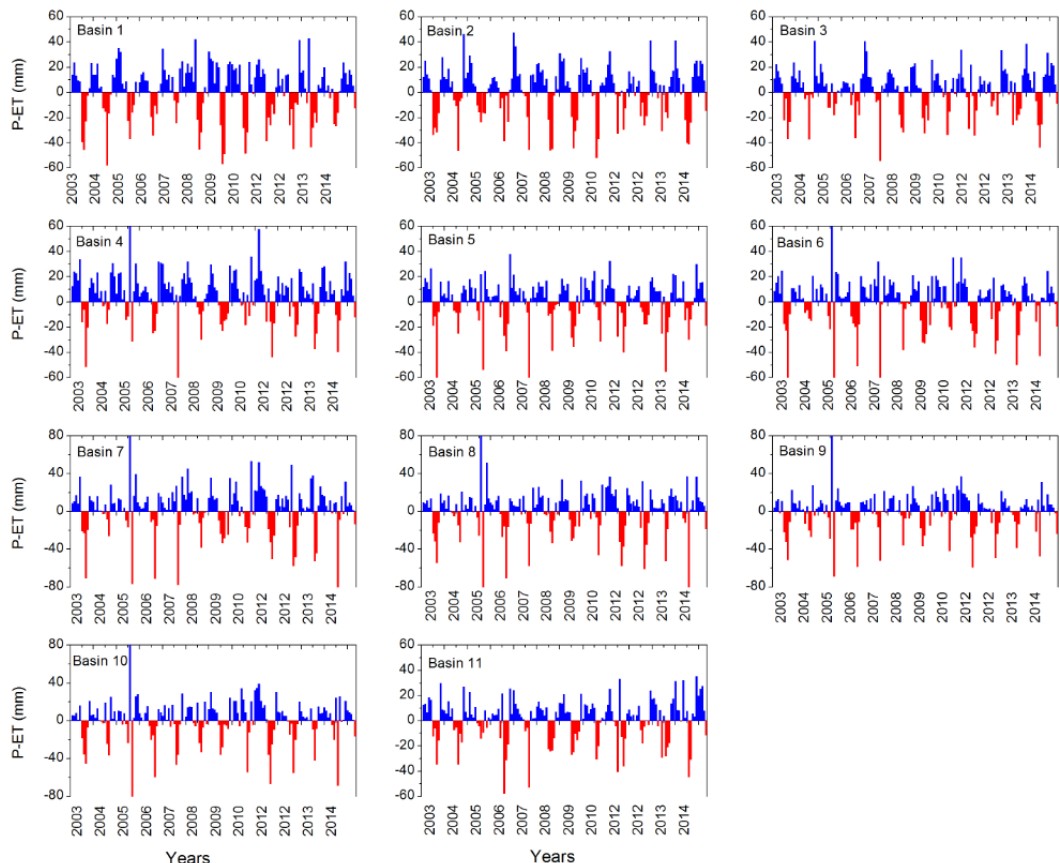

**Figure 9: Basin-wise time-series of P – ET values. Positive values are shown in blue colour and negative values are shown in red colour**

**Table 1: Details of the river basins used (within Alberta only), number of wells used, precipitation and GWSA_obs trends (statistically significant (*p value* < 0.01) trend estimates are shown in bold font)**

| Basin ID | Basin name | Ocean | Basin Area (m²) | Number of wells | Precipitation (mm/year) | GWSA_obs trends (cm/year) |
|---|---|---|---|---|---|---|
| 1 | Hay River Basin | Arctic Ocean | 66196942347 | 3 | 401 | **1.22** |
| 2 | Peace River Basin | Arctic Ocean | 213025952509 | 15 | 429 | -0.19 |





| 3 | Athabasca River Basin | Arctic Ocean | 144499671762 | 8 | 508 | **-0.19** |
| 4 | North Saskatchewan River | Hudson Bay | 57046775461 | 21 | 573 | **0.21** |
| 5 | Battle River Basin | Hudson Bay | 36561280700 | 28 | 424 | **0.43** |
| 6 | Red Deer River Basin | Hudson Bay | 50024664775 | 21 | 425 | **0.12** |
| 7 | Bow River Basin | Hudson Bay | 25639800168 | 15 | 546 | -0.04 |
| 8 | Oldman River Basin | Hudson Bay | 27023265616 | 10 | 486 | -0.08 |
| 9 | South Saskatchewan River Basin | Hudson Bay | 13504374212 | 6 | 334 | **-0.10** |
| 10 | Milk River Basin | Hudson Bay | 11833516877 | 9 | 353 | **0.52** |
| 11 | Beaver River Basin | Hudson Bay | 16904014071 | 21 | 469 | **0.24** |