# Peer review of "Estimating long-term groundwater storage and its controlling factors in Alberta, Canada"

_Hydrology and Earth System Sciences, 2018_

## Referee Comment (RC1) · Anonymous Referee #1 · 28 Jun 2018

General comments

The study presents an interesting use of gravity based remote sensing data (GRACE) for monitoring of groundwater resources in Alberta region and comparing it to available in situ monitoring well data. It is mostly nicely structured and written that the study is easy to follow for the reader. However, there are issues especially concerning the use of the data and the methods that should be revised thoroughly to enhance the quality of the manuscript.

Specific comments:

First main issue is how the in situ data is used. Authors mention in the abstract and in the text that the unconfined and unconfined aquifer monitoring wells are separated

from the in situ data and different approach for groundwater storage change has been used (equation 1 & 2). This is good as the well reading from confined aquifer compared to unconfined aquifer tell a different information on the aquifer storage. However, this separation of the data does not show in results or in discussion. In addition, this connects to the second issue of the manuscript. You have not given any information where on the studied catchments the wells are situated. As there is no spatial data for the wells or the information how the confined and unconfined aquifers are presented in each catchment, it is rather hard to say how representative the in situ data is for a specific catchment where you have the satellite data calculated and compared.

For example: the basin 7 in situ data and GRACE data do not seem to correlate. You have 15 wells in this catchment, but are these e.g. situated in one aquifer? Are they all unconfined aquifers? The average well data in figure 3 might indicate a strong annual snowmelt impact to the groundwater level in basin 7 average groundwater levels. This would happen in unconfined aquifers in snow dominated region (see comment on snow melt below). With the information given in the manuscript this cannot be confirmed or discussed in detail.

How deep aquifers the wells are monitoring? If the screening zone is for a deeper, confined aquifer, how much a yearly recharge impacts this aquifer? All in all, it would be beneficial to present more in detail how the monitoring wells are presenting the prevailing aquifer conditions in different catchments.

And concerning the methods used: the smallest catchment size (or part of the catchment studied) in this manuscript is Milk basin with 11834 km2. In total, the size in three of the catchments is smaller than 20000 km2. Is the size of the catchments a problem for the GRACE data methods used or does it cause uncertainty? This issue is previously discussed e.g. in Wishvakarma et al. 2017 for different GRACE approach.

Authors have studied how the precipitation is connected to the GWSA (chaper 3.5). However, role of snow is not discussed in detail. In many northern areas the snow melt

can be the driving factor for the groundwater storage recharge. Same goes to large areas in Alberta. As during the winter months the precipitation accumulates in snowpack and then usually melts in a short period, it would be more beneficial to compare warm period precipitation and winter time conditions (<0 degree C) separately, or take the snow water equivalent from GLDAS and add this to your analyses. With the straight comparison between monthly precipitation and GWSA a large portion of the yearly hydrological dynamics is missing. Authors have tested different approach in chapter 3.6., but this approach does not takes into account in detail the snow accumulation and snow melt.

Detailed comments:

Use of abbreviations: the text does not follow good order of abbreviations. E.g. in abstract in line 17 you present GWSAobs first time without explanation. And in line 19 has GWsat two times which mixes reader of the abstract. Same continues in text. E.g. in page 2 line 34 GWS is presented first time without explanation.

Page 2, line 20: extra comma

Page 2, line 9, space after point

Page 3, lines 17-18: sentence structure

Page 7: the two equations have a wrong number

Page 7, line 20: repetition from previous sentence

Page 10, line 6: 470 wells were monitored but 157 were used (page 4, lines 13-15) Is Figure 8 is not presented in the text.

---

## Short Comment (SC1) · 28 Jun 2018

The work by Bhanja et al. presents a study where GRACE observational products are compared with monitored and estimated groundwater storage changes in Alberta, Canada. The study shows that GRACE data can be used to understand groundwater storage changes and responses. As such, the results are important and the study of broad international interest. However, the manuscript is poorly organized and several sections are poorly written.

Main comments: The title promises too much. A more specific title with the focus on Alberta as case study would be more appropriate considering the content of the work. Also, is Alberta really a cold region or temperate region?

Generally, the role of snow accumulation and melt is not well discussed or included in the work. Certainly, this must be a main reason for seasonal changes in water storage in regions with winter and snow (4 seasons). Improve e.g. section 3.1. on this issue. Also on line 29, page 8, the statement on precipitation is a bit odd for cold climate (correct to snow).

I find it surprising that the global scale hydrological modelling used to estimate recharge has not explained in the methods at all (only shortly in section 3.5). A sub-section is needed on this under section 2 including aspects of uncertainty. Revise section 3.6 to focus on results of the modelling.

Detailed comments:

Several sections are poorly organized such as: • abstract: the 4 fist lines are too general. Provide 1 line as intro. • introduction: delete the first 2 paragraphs which are really poor in content (lines 1-18), and split the 3nd paragraph into 2-4 sub section on lines e.g. 23, 28 • the third objective is not presented as a number (bullet point) similar to the other sub-objectives. Why not?

In section 2, on lines 12-20, some information is provided about the aquifers. A map of the aquifers of Alberta could be useful. More importantly, how are the aquifers split into confined, semiconfined and unconfined?

In section 2.6, the equations 4-8 are general knowledge and should be deleted.

I feel that more information is needed on the comparison of GRACE MS and SH is needed in section 3.1.

Combine section 3.3-3.4. Also provide a meaningful title! RMSE etc. is not a good choice of title. Provide the result or outcome in the title.

The section 3.6. on assumptions is quite odd. Focus perhaps on uncertainty or delete the section, or put it into section 2.

The "conclusions" section 4 is well written.

---

## Author Comment (AC1) · 4 Aug 2018

*"Estimating long-term groundwater storage and its controlling factors in Alberta, Canada"*, by *Soumendra N. Bhanja, Xiaokun Zhang, Junye Wang*

**Reviewer #1**:

**General comments:** The study presents an interesting use of gravity based remote sensing data (GRACE) for monitoring of groundwater resources in Alberta region and comparing it to available in situ monitoring well data. It is mostly nicely structured and written that the study is easy to follow for the reader. However, there are issues especially concerning the use of the data and the methods that should be revised thoroughly to enhance the quality of the manuscript.

Reply: We would like to thank the reviewer for his/her interest in our work and also for careful consideration of the manuscript. We have addressed all of his/her concerns in the revised manuscript.

**Rev 1. Comment 1:** First main issue is how the in situ data is used. Authors mention in the abstract and in the text that the unconfined and unconfined aquifer monitoring wells are separated from the in situ data and different approach for groundwater storage change has been used (equation 1 & 2). This is good as the well reading from confined aquifer compared to unconfined aquifer tell a different information on the aquifer storage. However, this separation of the data does not show in results or in discussion. In addition, this connects to the second issue of the manuscript. You have not given any information where on the studied catchments the wells are situated. As there is no spatial data for the wells or the information how the confined and unconfined aquifers are presented in each catchment, it is rather hard to say how representative the in situ data is for a specific catchment where you have the satellite data calculated and compared.

For example: the basin 7 in situ data and GRACE data do not seem to correlate. You have 15 wells in this catchment, but are these e.g. situated in one aquifer? Are they all unconfined aquifers? The average well data in figure 3 might indicate a strong annual snowmelt impact to the groundwater level in basin 7 average groundwater levels. This would happen in unconfined aquifers in snow dominated region (see comment on snow melt below). With the information given in the manuscript this cannot be confirmed or discussed in detail.

Reply: We would like to thank the reviewer for raising this concern. We fully agreed with him. We have added Figs 1 d-e and Table S2 to show the spatial distributions of both confined and unconfined wells in each catchment. We have added two paragraphs within Section 3.1 to discuss effects of snowmelt and aquifer types and provided a figure in supplementary information.

Table S2: Basin-wide distribution of wells screened in different types of aquifers

| Basin | Unconfined | Semi-confined | Confined | Unclassified | Total |
|---|---|---|---|---|---|

| ID | | | | | |
|---|---|---|---|---|---|
| 1 | | | | 3 | **3** |
| 2 | 6 | | 7 | 2 | **15** |
| 3 | 2 | | 6 | | **8** |
| 4 | 5 | 2 | 14 | | **21** |
| 5 | 3 | 6 | 19 | | **28** |
| 6 | 1 | 3 | 16 | 1 | **21** |
| 7 | 3 | 1 | 4 | 7 | **15** |
| 8 | 2 | 1 | 6 | 1 | **10** |
| 9 | | 1 | 4 | 1 | **6** |
| 10 | 1 | 1 | 7 | | **9** |
| 11 | 1 | 2 | 17 | 1 | **21** |
| **Total** | **24** | **17** | **100** | **16** | **157** |

[Figure]

Figure 1: Major river basins in Alberta, (a) full basin extent; (b) Alberta only; (c) dominant land cover types; (d) aquifer types represented through the studied wells; (e) depth of wells screened in Alberta, overlaid by basin boundaries

"*Out of the 157 measurement locations used in the study, 24 are located in unconfined aquifers, 17 are located within semi-confined aquifers, 100 are located within confined aquifers and 16 are unclassified (Figure 1d). The screen depth of the wells varies from 6 m to 220 m (Figure 1e).*" [**Page: 4; Lines:** 7-10]

We added two paragraphs to discuss snowmelt impact and different types of aquifers.

"*Another important factor influencing groundwater recharge as well as the groundwater storage, is the snowmelt processes prevailing in cold regions during the onset of spring-summer. The river basins have been receiving substantial amount of snowfall during winter months (Figure 3). This leads to snow accumulation in the region. At the end of winter season, snowmelt processes are majorly accounting for our observation of increasing GWSA in April onwards (Figure 3). The observation is in line with the observations from the earlier studies conducted within the study region (Hayashi and Farrow, 2014; Hood and Hayashi, 2015). Comparatively higher rates of precipitation during summer months and the snowmelt during the start of the summer season, are the major processes responsible for the observation of higher GWSA during summertime at the entire study region (Figure 3).*" [**Page: 8; Lines:** 4-11]

*"GWSA$_{obs}$ values from the unconfined aquifers reflect higher magnitude than that in the confined aquifers (Figure S1). This is because of the intrinsic property of the different types of aquifers. For instance, dewatering from the saturated zone during a pumping event, is mainly responsible for the release of water in unconfined aquifer (Alley et al., 1999). On the other hand, a net decrease in groundwater potential and associated reduction in water pressure have been occurred during a pumping event in a confined aquifer. The indigenous water expands slightly due to the decrease in water pressure, leading to slight compression in the aquifer material (Alley et al., 1999). This can explain why the groundwater storage change in the confined aquifers are comparatively lower than that in the unconfined aquifers."* [**Page: 8; Lines:** 12-19]

[Figure]

[Figure]

Figure S1: Histogram of GWSA estimates from unconfined and confined aquifers

**Rev 1. Comment 2:** How deep aquifers the wells are monitoring? If the screening zone is for a deeper, confined aquifer, how much a yearly recharge impacts this aquifer? All in all, it would be beneficial to present more in detail how the monitoring wells are presenting the prevailing aquifer conditions in different catchments.

Reply: We would like to thank the reviewer for his/her suggestion. We have provided more details on well depth and types of aquifers encountered in the revised version of the manuscript. Please see our answer to **Rev 1. Comment 1**. Recharge impact on groundwater storage in confined aquifer is a complex issue to deal with, this is beyond the scope of this manuscript at present. Here, we are not dealing with the absolute storage but estimating the storage anomaly (that is the deviation of storage from a mean value). If the confined aquifer recharge is constant over the years, it will be cancelled out by computing storage anomaly.

**Rev 1. Comment 3:** And concerning the methods used: the smallest catchment size (or part of the catchment studied) in this manuscript is Milk basin with 11834 km2. In total, the size in three of the catchments is smaller than 20000 km2. Is the size of the catchments a problem for the GRACE data methods used or does it cause uncertainty? This issue is previously discussed e.g. in Wishvakarma et al. 2017 for different GRACE approach.

Reply: We thank the reviewer for his/her suggestion. We agree that the use of GRACE data is not always appropriate for smaller basins. We have discussed these issues in Section 2.7 Assumptions and limitations.

*"We have shown the satellite-based estimates for all of the basins, however, users should be cautious to use GRACE data in the smallest basins. This is because GRACE's native resolution could not allow users to directly use the data for smaller basins. Other processes, such as, the use of GRACE and integrated land surface model's operation could make the data available to use for smaller basins (Landerer and Swenson, 2012; Watkins et al., 2015). Data processing methods Proposed by Dutt Vishwakarma et al. (2016) could be used to make the data available for smaller basins with GRACE-SH products."* [**Page: 7; Lines:** 15-20]

**Rev 1. Comment 4:** Authors have studied how the precipitation is connected to the GWSA (chaper 3.5). However, role of snow is not discussed in detail. In many northern areas the snow melt can be the driving factor for the groundwater storage recharge. Same goes to large areas in Alberta. As during the winter months the precipitation accumulates in snowpack and then usually melts in a short period, it would be more beneficial to compare warm period precipitation and winter time conditions (<0 degree C) separately, or take the snow water equivalent from GLDAS and add this to your analyses. With the straight comparison between monthly precipitation and GWSA a large portion of the yearly hydrological dynamics is missing. Authors have tested different approach in chapter 3.6., but this approach does not takes into account in detail the snow accumulation and snow melt.

Reply: We would like to thank the reviewer for his/her concern. We have now included the analyses of snowmelt and its influence on GWSA. We have modified the Figure 8 and include the combined data of rainfall and snowmelt along with the precipitation and GWSA. We have modified the Section 3.3 as:

*"In general, precipitation is the major controlling factor for variations in water storage (Scanlon et al., 2012). In this study, we have observed that GWSA values are not directly influenced by the precipitation pattern in some of the basins (Figure 8). The HP trend analysis shows a good match of $GWSA_{obs}$ with precipitation in basins 1 and 10 only (Figure 8, Table S5). $GWSA_{obs}$ trends are not following precipitation pattern in other basins (Figure 8, Table S5). The cross-correlation analysis between HP trends provide similar inferences (Table S5). In order to investigate the relationship with more detail, the Granger causality analyses (Granger, 1988) were performed with order 1 (insignificant results were found when other orders were used). Results show precipitation significantly (p value <0.01) causes $GWSA_{obs}$ in 4 of the 11 studied basins, basin 1, 5, 7 and 11. The results were found to be insignificant or even negatively correlated in other basins (Table S5).*

*A part of the precipitation, in particular, snowfall has little influence in modulating the groundwater storage, unless it is converted to snowmelt water. Therefore, we have studied the combined influence of rainfall and snowmelt water on $GWSA_{obs}$. Here, the rainfall and the snowmelt water data are retrieved from the three LSMs (CLM, VIC and Noah) in GLDAS archive and used in combination. Good match between rainfall and snowmelt water, and $GWSA_{obs}$ have been obtained in basins 1 and 11. Cross-correlation analyses indicate similar*

*inference (Table S6). Granger causality analyses (order 1) show the combined effect of rainfall and snowmelt water significantly causes GWSA$_{obs}$ in 6 basins: 1, 2, 5, 7, 9 and 11 respectively. This implies that other factors, such as domestic and industrial water withdrawal etc., play major roles in influencing the GWSA in other basins."* [**Pages: 9-10; Lines:** 19-2]

[Figure]

**Figure 8: Basin-wide time-series of HP filter data for in situ GWSA (OBS, red squares), precipitation data (green circles) and rainfall+snowmelt data (blue circles). Pearson's correlation coefficient (r) values are provided in in-set and statistically significant (*p value* < 0.01) values are shown in bold font. r$_p$ and r$_{rs}$ indicate correlation between GWSA, and precipitation and rainfall+snowmelt, respectively**

Also included these in abstract and conclusions.

*"A combination of rainfall and snowmelt positively influence the GWSA$_{obs}$ in 6 basins."* [**Page: 1; Lines:** 19-20]

*"A combination of rainfall and snowmelt water causes significant GWSA variations in 6 basins, indicating prevalence of other factors for influencing GWSA in the remaining basins."* [**Page: 11; Lines:** 2-4]

We have also added cross-correlation analyses details in Table S6 between rainfall+snowmelt and the $GWSA_{obs}$.

Table S6: Correlation analysis between Hodrick-Prescott trend of rainfall+snowmelt and $GWSA_{obs}$ (no lag, 1 month lag and 2 months lag)

| Basin id | R No lag | R 1 month lag | R 2 months lag |
|---|---|---|---|
| 1 | 0.69 | 0.72 | 0.75 |
| 2 | 0.50 | 0.47 | 0.43 |
| 3 | 0.00 | -0.02 | -0.03 |
| 4 | -0.01 | 0.02 | 0.05 |
| 5 | 0.02 | 0.06 | 0.10 |
| 6 | 0.10 | 0.13 | 0.16 |
| 7 | 0.77 | 0.76 | 0.74 |
| 8 | 0.08 | 0.06 | 0.05 |
| 9 | -0.18 | -0.19 | -0.20 |
| 10 | 0.33 | 0.37 | 0.40 |
| 11 | 0.77 | 0.77 | 0.76 |

In the revised version of the manuscript, we have also discussed the snowmelt issues in the Result and Discussions Section 3.1.

"*Another important factor influencing groundwater recharge as well as the groundwater storage, is the snowmelt processes prevailing in cold regions during the onset of spring-summer. The river basins have been receiving substantial amount of snowfall during winter months (Figure 3). This leads to snow accumulation in the region. At the end of winter season, snowmelt processes are majorly accounting for our observation of increasing GWSA in April onwards (Figure 3). The observation is in line with the observations from the earlier studies conducted within the study region (Hayashi and Farrow, 2014; Hood and Hayashi, 2015). Comparatively higher rates of precipitation during summer months and the snowmelt during the start of the summer season, are the major processes responsible for the observation of higher GWSA during summertime at the entire study region (Figure 3).*" [**Page: 8; Lines:** 4-11]

**Detailed comments:**

**Rev 1. Comment 5:** Use of abbreviations: the text does not follow good order of abbreviations. E.g. in abstract in line 17 you present GWSAobs first time without explanation. And in line 19 has GWsat two times which mixes reader of the abstract. Same continues in text. E.g. in page 2 line 34 GWS is presented first time without explanation.

Reply: We would like to thank the reviewer for his/her careful observation. We have modified the sentences as suggested by the reviewer.

"*Storage coefficients for the individual wells were incorporated to compute the monthly in situ groundwater storage (GWSA$_{obs}$).*" [**Page: 1; Lines:** 13-14]

"*They used ground water levels at 36 wells, mostly confined to the southern Alberta region, and were correlated with both the GRACE total water storage (TWS) and groundwater storage (GWS) variations.*" [**Page: 2; Lines:** 14-16]

**Rev 1. Comment 6:** Page 2, line 20: extra comma

Reply: Thanks for the observation. The comma has been deleted.

**Rev 1. Comment 7:** Page 2, line 9, space after point

Reply: Thanks for the observation. A space is given after the point in the revised version.

**Rev 1. Comment 8:** Page 3, lines 17-18: sentence structure

Reply: Following the reviewer's suggestion, we have modified the sentences in the revised version.

"*To find the role of natural hydrological components (e.g. precipitation, snowmelt, evapotranspiration) for influencing groundwater storage variations. We have also studied long-term groundwater recharge trends from a global-scale hydrological model for inferring long-term variabilities in groundwater recharge rates.*" [**Page: 3; Lines:** 4-6]

**Rev 1. Comment 9:** Page 7: the two equations have a wrong number

Reply: We would like to thank the reviewer for his/her careful consideration. The equation numbers are modified in the revised version.

**Rev 1. Comment 10:** Page 7, line 20: repetition from previous sentence

Reply: Following the reviewer's suggestion, we have modified the sentence.

*"Basin 3 has been subjected to the highest amount of licensed groundwater withdrawal allocation in Alberta (basin 3 accounts for 39% of the total groundwater usage in Alberta)."* [**Page: 7; Lines:** 28-30]

**Rev 1. Comment 11:** Page 10, line 6: 470 wells were monitored but 157 were used (page 4, lines 13-15) Is Figure 8 is not presented in the text.

Reply: We would like to thank the reviewer for his/her suggestions. We have modified the sentence in conclusion, reflecting the number of wells use for final analyses.

*"A network of 157 daily groundwater monitoring wells was used to compute groundwater storage anomalies (GWSA) in 11 major river basins in Alberta, Canada between January 2003 and April 2015."* [**Page: 10; Lines:** 22-23]

We have referred the Figure 8 in text within Section 3.3.

*"In general, precipitation is the major controlling factor for variations in water storage (Scanlon et al., 2012). In this study, we have observed that GWSA values are not directly influenced by the precipitation pattern in some of the basins (Figure 8). The HP trend analysis shows a good match of $GWSA_{obs}$ with precipitation in basins 1 and 10 only (Figure 8, Table S5). $GWSA_{obs}$ trends are not following precipitation pattern in other basins (Figure 8, Table S5). The cross-correlation analysis between HP trends provide similar inferences (Table S5). In order to investigate the relationship with more detail, the Granger causality analyses (Granger, 1988) were performed with order 1 (insignificant results were found when other orders were used). Results show precipitation significantly ($p$ value $<0.01$) causes $GWSA_{obs}$ in 4 of the 11 studied basins, basin 1, 5, 7 and 11. The results were found to be insignificant or even negatively correlated in other basins (Table S5).*

*A part of the precipitation, in particular, snowfall has little influence in modulating the groundwater storage, unless it is converted to snowmelt water. Therefore, we have studied the combined influence of rainfall and snowmelt water on $GWSA_{obs}$. Here, the rainfall and the snowmelt water data are retrieved from the three LSMs (CLM, VIC and Noah) in GLDAS archive and used in combination. Good match between rainfall and snowmelt water, and $GWSA_{obs}$ have been obtained in basins 1 and 11. Cross-correlation analyses indicate similar inference (Table S6). Granger causality analyses (order 1) show the combined effect of rainfall and snowmelt water significantly causes $GWSA_{obs}$ in 6 basins: 1, 2, 5, 7, 9 and 11 respectively. This implies that other factors, such as domestic and industrial water withdrawal etc., play major roles in influencing the GWSA in other basins."* [**Pages: 9-10; Lines:** 19-2]

---

## Author Comment (AC2) · 4 Aug 2018

*"Estimating long-term groundwater storage and its controlling factors in Alberta, Canada"*, by *Soumendra N. Bhanja, Xiaokun Zhang, Junye Wang*

**Prof. B. Klöve's comments:**

**SC #1 General comments:** The work by Bhanja et al. presents a study where GRACE observational products are compared with monitored and estimated groundwater storage changes in Alberta, Canada. The study shows that GRACE data can be used to understand groundwater storage changes and responses. As such, the results are important and the study of broad international interest. However, the manuscript is poorly organized and several sections are poorly written.

Reply: We sincerely thank Prof. Klöve for his interest in our work. We have thoroughly revised the manuscript to take into consideration your concerns and comments. We have re-organized and rewritten several sections, insert new figures, tables and discussions, which we believe, have improved the manuscript to a great respect.

**SC1 Comment 1:** The title promises too much. A more specific title with the focus on Alberta as case study would be more appropriate considering the content of the work. Also, is Alberta really a cold region or temperate region?

Reply: Following Prof. Klöve's suggestion, we have modified the title to "*Estimating long-term groundwater storage and its controlling factors in Alberta, Canada*". In general, most parts of Alberta falls within cold climate region based on Koppen-Geiger classification. Please refer to Figure 6 in Peel et al. (2007).

**SC1 Comment 2:** Generally, the role of snow accumulation and melt is not well discussed or included in the work. Certainly, this must be a main reason for seasonal changes in water storage in regions with winter and snow (4 seasons). Improve e.g. section 3.1. on this issue. Also on line 29, page 8, the statement on precipitation is a bit odd for cold climate (correct to snow).

Reply: We would like to thank Prof. Klöve for raising this concern. We agree with him that snowmelt is a main factor in groundwater storage in cold region. We have now included the analyses of snowmelt and its influence on GWSA. We have modified the Figure 8 and included the combined data of rainfall and snowmelt along with the precipitation and GWSA. We have modified the subsection 3.3 as:

*"In general, precipitation is the major controlling factor for variations in water storage (Scanlon et al., 2012). In this study, we have observed that GWSA values are not directly influenced by the precipitation pattern in some of the basins (Figure 8). The HP trend analysis shows a good match of $GWSA_{obs}$ with precipitation in basins 1 and 10 only (Figure 8, Table S5). $GWSA_{obs}$ trends are not following precipitation pattern in other basins (Figure 8, Table S5). The cross-correlation analysis between HP trends provide similar inferences (Table S5). In order to*

*investigate the relationship with more detail, the Granger causality analyses (Granger, 1988) were performed with order 1 (insignificant results were found when other orders were used). Results show precipitation significantly (p value <0.01) causes GWSA$_{obs}$ in 4 of the 11 studied basins, basin 1, 5, 7 and 11. The results were found to be insignificant or even negatively correlated in other basins (Table S5).*

*A part of the precipitation, in particular, snowfall has little influence in modulating the groundwater storage, unless it is converted to snowmelt water. Therefore, we have studied the combined influence of rainfall and snowmelt water on GWSA$_{obs}$. Here, the rainfall and the snowmelt water data are retrieved from the three LSMs (CLM, VIC and Noah) in GLDAS archive and used in combination. Good match between rainfall and snowmelt water, and GWSA$_{obs}$ have been obtained in basins 1 and 11. Cross-correlation analyses indicate similar inference (Table S6). Granger causality analyses (order 1) show the combined effect of rainfall and snowmelt water significantly causes GWSA$_{obs}$ in 6 basins: 1, 2, 5, 7, 9 and 11 respectively. This implies that other factors, such as domestic and industrial water withdrawal etc., play major roles in influencing the GWSA in other basins.*" [**Pages: 9-10; Lines:** 19-2]

[Figure]

**Figure 8: Basin-wide time-series of HP filter data for in situ GWSA (OBS, red squares), precipitation data (green circles) and rainfall+snowmelt data (blue circles). Pearson's**

**correlation coefficient (r) values are provided in in-set and statistically significant (*p value* < 0.01) values are shown in bold font. $r_p$ and $r_{rs}$ indicate correlation between GWSA, and precipitation and rainfall+snowmelt, respectively**

Also included these in abstract and conclusions.

"*A combination of rainfall and snowmelt positively influence the GWSA$_{obs}$ in 6 basins.*"[**Page: 1; Lines:** 19-20]

"*A combination of rainfall and snowmelt water causes significant GWSA variations in 6 basins, indicating prevalence of other factors for influencing GWSA in the remaining basins.*" [**Page: 11; Lines:** 2-4]

We have also added cross-correlation analyses details in Table S6 between rainfall+snowmelt and the GWSA$_{obs}$.

Table S6: Correlation analysis between Hodrick-Prescott trend of rainfall+snowmelt and GWSA$_{obs}$ (no lag, 1 month lag and 2 months lag)

| Basin id | R No lag | R 1 month lag | R 2 months lag |
|---|---|---|---|
| 1 | 0.69 | 0.72 | 0.75 |
| 2 | 0.50 | 0.47 | 0.43 |
| 3 | 0.00 | -0.02 | -0.03 |
| 4 | -0.01 | 0.02 | 0.05 |
| 5 | 0.02 | 0.06 | 0.10 |
| 6 | 0.10 | 0.13 | 0.16 |
| 7 | 0.77 | 0.76 | 0.74 |
| 8 | 0.08 | 0.06 | 0.05 |
| 9 | -0.18 | -0.19 | -0.20 |
| 10 | 0.33 | 0.37 | 0.40 |
| 11 | 0.77 | 0.77 | 0.76 |

In the revised version of the manuscript, we have also discussed the snowmelt issues in the Result and Discussions Section 3.1.

"*Another important factor influencing groundwater recharge as well as the groundwater storage, is the snowmelt processes prevailing in cold regions during the onset of spring-summer. The river basins have been receiving substantial amount of snowfall during winter months (Figure 3). This leads to snow accumulation in the region. At the end of winter season, snowmelt processes are majorly accounting for our observation of increasing GWSA in April onwards (Figure 3). The observation is in line with the observations from the earlier studies conducted within the study region (Hayashi and Farrow, 2014; Hood and Hayashi, 2015). Comparatively higher rates of precipitation during summer months and the snowmelt during the start of the summer season, are the major processes responsible for the observation of higher GWSA during summertime at the entire study region (Figure 3).*" [**Page: 8; Lines:** 4-11]

**SC1 Comment 3:** I find it surprising that the global scale hydrological modelling used to estimate recharge has not explained in the methods at all (only shortly in section 3.5). A sub-section is needed on this under section 2 including aspects of uncertainty. Revise section 3.6 to focus on results of the modelling.

Reply: Following Prof. Klöve's suggestion, we have included subsection 2.6 for description of the model generated groundwater recharge output including uncertainties. Section 3.4 (earlier 3.6) has now included only the results and discussions related to the modelled recharge.

We used global scale hydrological modelling output to discuss about the patterns and trends of the groundwater prevailing over the region. There are no direct measurement for groundwater recharge is available.

"*We have studied the long-term (1960-2009) groundwater recharge occurrence from the global-scale model output because of unavailability of direct groundwater recharge measurement in the region.*" [**Page: 10; Lines:** 14-15]

"*2.6 Groundwater recharge from global-scale hydrological model*

*In order to find the historical groundwater recharge pattern, we used a global-scale hydrological model, WaterGAP (version 2.2) (Doll et al., 2014) to estimate long-term groundwater recharge data (1960-2009). The WaterGAP simulates global-scale water storage and transport including human water use and groundwater recharge from surface water bodies at $0.5^0 \times 0.5^0$ resolution (Doll et al., 2014). Water withdrawal from both groundwater and surface water have also been considered. We used a combination of diffuse groundwater recharge and recharge from the surface water bodies, which we termed as "total groundwater recharge". As the WaterGAP simulation consider simple water balance approach for groundwater recharge estimation, uncertainties may arise as a function of groundwater table gradient (Doll et al., 2014). Furthermore, increasing groundwater recharge from surface water*

*bodies as a function of groundwater withdrawal, has not been considered here (Doll et al., 2014). More information on model processes, data used and other details can be found in Doll et al. (2014).*" [**Page: 6; Lines:** 16-24]

**Detailed comments:**

**SC1 Comment 4:** Several sections are poorly organized such as: â˘A ´c abstract: the 4 fist lines are too general. Provide 1 line as intro. â˘A ´c introduction: delete the first 2 paragraphs which are really poor in content (lines 1-18), and split the 3nd paragraph into 2-4 sub section on lines e.g. 23, 28 â˘A ´c the third objective is not presented as a number (bullet point) similar to the other sub-objectives. Why not?

Reply: We thank Prof. Klöve for his suggestion. We have modified the sections in the revised version of the manuscript. We have deleted two introductory sentences from Abstract.

Following your suggestion, we have deleted the initial paragraphs of Introduction section and divided the third paragraph into three paragraphs.

We would like to thank Prof. Klöve for his careful observation. The third objective is now numbered as 3.

[revised manuscript text omitted]

**SC1 Comment 5:** In section 2, on lines 12-20, some information is provided about the aquifers. A map of the aquifers of Alberta could be useful. More importantly, how are the aquifers split into confined, semiconfined and unconfined?

Reply: We would like to thank Prof. Klöve for his concern. We have added two new figures within Figure 1, indicating the types of aquifer encountered and the screen depth of the wells. We have also discussed this issue in text and provided a figure and a table in supplementary information. Aquifer map development is out of the scope of this study, however, it can be found in Lemay and Guha (2009).

Table S2: Basin-wide distribution of wells screened in different types of aquifers

| Basin ID | Unconfined | Semi-confined | Confined | Unclassified | Total |
|---|---|---|---|---|---|
| 1 | | | | 3 | **3** |

| | | | | | |
|---|---|---|---|---|---|
| 2 | 6 | | 7 | 2 | **15** |
| 3 | 2 | | 6 | | **8** |
| 4 | 5 | 2 | 14 | | **21** |
| 5 | 3 | 6 | 19 | | **28** |
| 6 | 1 | 3 | 16 | 1 | **21** |
| 7 | 3 | 1 | 4 | 7 | **15** |
| 8 | 2 | 1 | 6 | 1 | **10** |
| 9 | | 1 | 4 | 1 | **6** |
| 10 | 1 | 1 | 7 | | **9** |
| 11 | 1 | 2 | 17 | 1 | **21** |
| **Total** | **24** | **17** | **100** | **16** | **157** |

[Figure]

Figure 1: Major river basins in Alberta, (a) full basin extent; (b) Alberta only; (c) dominant land cover types; (d) aquifer types represented through the studied wells; (e) depth of wells screened in Alberta, overlaid by basin boundaries

*"Out of the 157 measurement locations used in the study, 24 are located in unconfined aquifers, 17 are located within semi-confined aquifers, 100 are located within confined aquifers and 16 are unclassified (Figure 1d). The screen depth of the wells varies from 6 m to 220 m (Figure 1e)."* [**Page: 4; Lines:** 7-10]

We added two paragraphs to discuss snowmelt impact and different types of aquifers.

*"Another important factor influencing groundwater recharge as well as the groundwater storage, is the snowmelt processes prevailing in cold regions during the onset of spring-summer. The river basins have been receiving substantial amount of snowfall during winter months (Figure 3). This leads to snow accumulation in the region. At the end of winter season, snowmelt processes are majorly accounting for our observation of increasing GWSA in April onwards (Figure 3). The observation is in line with the observations from the earlier studies conducted within the study region (Hayashi and Farrow, 2014; Hood and Hayashi, 2015). Comparatively higher rates of precipitation during summer months and the snowmelt during the start of the summer season, are the major processes responsible for the observation of higher GWSA during summertime at the entire study region (Figure 3)."* [**Page: 8; Lines:** 4-11]

*"$GWSA_{obs}$ values from the unconfined aquifers reflect higher magnitude than that in the confined aquifers (Figure S1). This is because of the intrinsic property of the different types of aquifers. For instance, dewatering from the saturated zone during a pumping event, is mainly responsible for the release of water in unconfined aquifer (Alley et al., 1999). On the other hand, a net decrease in groundwater potential and associated reduction in water pressure have been occurred during a pumping event in a confined aquifer. The indigenous water expands slightly due to the decrease in water pressure, leading to slight compression in the aquifer material (Alley et al., 1999). This can explain why the groundwater storage change in the confined aquifers are comparatively lower than that in the unconfined aquifers."* [**Page: 8; Lines:** 12-19]

[Figure]

Figure S1: Histogram of GWSA estimates from unconfined and confined aquifers

**SC1 Comment 6:** In section 2.6, the equations 4-8 are general knowledge and should be deleted.

Reply: Following the suggestion, we have moved the Equations 4-8 to Supplementary information in the revised manuscript.

**SC1 Comment 7:** I feel that more information is needed on the comparison of GRACE MS and SH is needed in section 3.1.

Reply: Based on Figure 3 only, it is found that the GRACE MS and SH estimates match each other. Detailed information on comparison of GRACE MS and SH products are provided in section 3.2. We indicated this at the end of section 3.1.

"*In general, the magnitude of the GWSA$_{sat}$ compares well with that of the GWSA$_{obs}$ (Figure 3).*" [**Page: 8; Line:** 21]

"*Overall, the two satellite based estimates are found to be closely matching with one another, detailed comparisons are provided in section 3.2.*" [**Page: 8; Lines:** 24-25]

**SC1 Comment 8:** Combine section 3.3-3.4. Also provide a meaningful title! RMSE etc. is not a good choice of title. Provide the result or outcome in the title.

Reply: We would like to thank Prof. Klöve for raising this concern. We have merged the three section 3.2-3.4 to make a single Section 3.2 "*Comparison between observed and satellite-based GWSA*". We have also modified the title of the section.

**SC1 Comment 9:** The section 3.6. on assumptions is quite odd. Focus perhaps on uncertainty or delete the section, or put it into section 2.

Reply: Following your suggestion, we have moved the Section 3.6 from a "*Results and Discussions*" section to "*Materials and Methods*" section. The new section number is 2.8.

**SC1 Comment 10:** The "conclusions" section 4 is well written.

Reply: We would like to thank Prof. Klöve for his appreciation.

---

## Author Response (AR3)

*"Estimating long-term groundwater storage and its controlling factors in Alberta, Canada"*, by *Soumendra N. Bhanja, Xiaokun Zhang, Junye Wang*

**General Comment from Authors and highlights of revision:**
We have responded to all of the reviewers' concerns in the previous version of the manuscript. A point-by-point responses is provided below.

**Reviewer #1**:

**General comments:** Authors have revised the text according to the reviewers comments and e.g. now the text takes more into account the snow conditions of the study area and is more detailed in the spatial distribution, depth and type of the monitoring wells. Some small remarks still need attention.

Reply: We would like to thank the reviewer for his/her careful consideration of the manuscript. We have addressed all of his/her concerns in the revised manuscript.

**Rev 1. Comment 1:** How the rain+snowmelt is defined in chapter 3.3.? Is the snowfall during wintertime separated from the rain-part? This should be clarified in the text.

Reply: In the study, we have not considered direct snowfall but rainfall amount along with the snowmelt values. We have clarified this in the text.

" *Therefore, we have studied the combined influence of rainfall and snowmelt water on GWSA$_{obs}$ without considering the direct snowfall amount. The rainfall and the snowmelt water data are retrieved from the three LSMs (CLM, VIC and Noah) in GLDAS archive and used in combination.*" [**Page: 9; Lines: 30-32**]

**Rev 1. Comment 2:** On page 10, line 1 You discuss in the results e.g. gw-use as one possible reason for GWSA change: "This implies that other factors, such as domestic and industrial water withdrawal etc., play major roles in influencing the GWSA in other basins." And on the next chapter 3.4 you continue with DIFF in the text and it seems that P-ET is reducing in 9 out of 11 subcatchments. This leaves reader bit puzzled, as you make an implication and then add one more aspect to the study. This would need a bit better connection between the chapters.

Reply: We would like to thank the reviewer for his/her suggestion. We have removed the last sentence of section 3.3 in order to show better connection between the two sections. The initial sentences of the revised section 3.4 are following:

"*Observation of a non-significant relationship of precipitation, snowmelt water and GWS anomalies in most of the basins indicated the influence of other factors controlling GWS. In this aspect, the natural water availability for terrestrial water components (i.e. groundwater, surface water, soil moisture, etc.) have been studied by delineating the difference (DIFF) between*

*precipitation (P) and evapotranspiration (ET) in another way, called the net precipitation flux (Syed et al., 2005; Rodell et al., 2015)."* [**Page: 10; Lines: 4-8**]

**Rev 1. Comment 3:** Also on more aspect of the chapter 3.4: is the declining P-ET due to climate change? What would be the estimate for P and ET in climate change models for Alberta?

Reply: Estimates of P - ET change due to climate change is out of the scope of this study. This is a potential topic for another research which can be addressed in future research. Based on the available data, we have shown the decrease of P - ET during the study period.

**Rev 1. Comment 4:** Page 4, line 4: "The data is screened for data continuity (at least 80% of the data are present in each location)". Clarify this. 80 % data present temporally?

Reply: Out of 470 available groundwater level monitoring locations, data recording intervals in approximately two third of the wells are intermittent during the study period. Therefore, we applied a filter of at least 80% of the temporal data availability at any locations for their further usage in our analyses. We have also clarified this in the text.

*"Out of 470 available groundwater level monitoring locations, data recording intervals in approximately two third of the wells are intermittent during the study period. Therefore, we applied a filter of at least 80% of the temporal data availability at any locations for their further usage in our analyses within the study period 2003-2015,, resulting in the use of GWL data from 157 measurement locations."* [**Page: 4; Lines: 4-7**]

**Rev 1. Comment 5:** Page 10, lines 11-13. Reader gets confused here. Table 1 shows GWSAobs variation but here you refer to Table 1 where DIFF results should be presented. Should the basin-wise DIFF values be found somewhere, where 9 out of 11 subcatchments have the negative values?

Reply: We would like to thank the reviewer for his/her concern. We have inserted the basin-wise DIFF trends i.e. P-ET trends in Table 1 in the revised version of the manuscript.

*" Table 1: Details of the river basins used (within Alberta only), number of wells used, precipitation, $GWSA_{obs}$ trends (statistically significant (p value< 0.01) trend estimates are shown in bold font) and P - ET trends in 2003-2015*

| Basin ID | Basin name | Ocean | Basin Area $(m^2)$ | Number of wells | Precipitation (mm/year) | $GWSA_{obs}$ trends (cm/year) | P - ET trends ($km^3$ in 2003- |
|---|---|---|---|---|---|---|---|

| | | | | | | | |
|---|---|---|---|---|---|---|---|
| **1** | Hay River Basin | Arctic Ocean | 66196942347 | 3 | 401 | **1.22** | -0.17 |
| **2** | Peace River Basin | Arctic Ocean | 213025952509 | 15 | 429 | -0.19 | -0.41 |
| **3** | Athabasca River Basin | Arctic Ocean | 144499671762 | 8 | 508 | **-0.19** | -0.17 |
| **4** | North Saskatchewan River | Hudson Bay | 57046775461 | 21 | 573 | **0.21** | -0.25 |
| **5** | Battle River Basin | Hudson Bay | 36561280700 | 28 | 424 | **0.43** | -0.06 |
| **6** | Red Deer River Basin | Hudson Bay | 50024664775 | 21 | 425 | **0.12** | -0.11 |
| **7** | Bow River Basin | Hudson Bay | 25639800168 | 15 | 546 | -0.04 | -0.08 |
| **8** | Oldman River Basin | Hudson Bay | 27023265616 | 10 | 486 | -0.08 | -0.03 |
| **9** | South Saskatchewan River Basin | Hudson Bay | 13504374212 | 6 | 334 | **-0.10** | -0.01 |
| **10** | Milk River Basin | Hudson Bay | 11833516877 | 9 | 353 | **0.52** | 0.00 |
| **11** | Beaver River Basin | Hudson Bay | 16904014071 | 21 | 469 | **0.24** | 0.01 |

"

**Prof. B. Klöve's comments:**

**SC1 Comment 1:** In Abstract, please consider revising of the first sentence (as it is too general). I recommend to link it to the title of your work and/or the key objectives.

Reply: Following Prof. Klöve's suggestion, we have modified the first sentence as "*Groundwater is one of the crucial natural resources for economic development and environmental sustainability particularly in Alberta, Canada.*". [**Page: 1; Lines: 9-10**]

**SC1 Comment 2:** I think spelling of the author "Doll" should be "Döll". Please check.

Reply: We have rectified the author's name as "Döll" in the revised manuscript.

[revised manuscript text omitted]